# Sustainable and Secure Transport: Achieving Environmental Impact Reductions by Optimizing Pallet-Package Strength Interactions during Transport

**Saewhan Kim** [1] , **Laszlo Horvath** [1] , **Jennifer D. Russell** [1,*] **and Jonghun Park** [2]

1 Department of Sustainable Biomaterials, Virginia Polytechnic Institute and State University, Blacksburg, VA 24061, USA; seabed94@vt.edu (S.K.); lhorvat@vt.edu (L.H.)

2 Graphic Communications Management at the Creative School, Toronto Metropolitan University, Toronto, ON M5B 2K3, Canada; jaypark@torontomu.ca

\* Correspondence: jdrussell@vt.edu; Tel.: +1-540-231-9516

**Abstract:** Increasing quantities of products are being transported across widely distributed supply networks; the sustainability of the packaging used to transport these goods, or unit loads, presents an area of potential concern. The most common type of unit load in the U.S. is wooden pallets supporting various configurations of stacked corrugated boxes. Research into unit load cost optimization revealed that increasing the stiffness of a pallet's top deck can significantly affect the strength of the assembled, stacked corrugated boxes and provides opportunities to reduce the board grade required for accompanying corrugated boxes. However, there remains a knowledge gap regarding the environmental implications of this type of unit load optimization method. To address this, we conducted a life cycle analysis (LCA) to investigate the environmental implications of optimizing a unit load using this method. The environmental impacts of paired (pallet and box) unit load design scenarios ($n = 108$) were investigated using varied wood species, pallet top deck thicknesses, corrugated boxes sizes, corrugated flutes, and board grades. Initial and optimized unit load scenarios ensured that the unit loads offered equivalent performance. LCA results indicate that optimizing the unit load can reduce environmental impacts by up to 23%, with benefits accruing across most impact categories primarily due to the reduction in corrugated material used. Ozone depletion, the exception, was mainly affected by the increase in the amount of required pallet materials. This study provides minimum required conditions as preliminary guidance for determining the usefulness of unit load specific analysis, and a sensitivity analysis confirmed these values remain unchanged even with different transportation distances. Through the unit load optimization method, this study demonstrates that an effective way to reduce the overall environmental impact and cost of transported unit loads involves increasing the stiffness of the top decks and reducing the corrugated board grade.

**Keywords:** packaging sustainability; distribution packaging; unit load; pallet; corrugated box

## 1. Introduction

Packaging is widely used to contain, protect, preserve, and transport goods. Besides these essential functions, end-of-life (EOL) scenarios for packaging have also become an area of growing interest. With the rise of concern about sustainability issues, the sustainability of packaging has also drawn tremendous attention due to resulting high volumes of waste generation across supply chains [1]. In 2018, about 28% (82 million tons) of municipal solid waste generated in the United States was reported as packaging-related materials [2]. As the importance of sustainability has increased, packaging has rapidly become recognized as an area that requires immediate attention by consumers, industry, and policymakers [3,4].

Distribution packaging plays a crucial role in the transportation of goods, ensuring the primary packaging, safety, efficiency, and cost-effectiveness in getting goods to wherever

consumers can easily access them. Eighty percent (80%) of distribution packaging in the U.S. takes the form of a 'unit load', which consists of the configured combination of a pallet, stacked packaging (e.g., corrugated boxes), and the material handling system [5,6]. As the unit load format accounts for a large share of the distribution packaging system, it is crucial to understand and evaluate the environmental impact of the unit load. However, previous studies exploring the environmental impacts of unit load have treated the two primary components, i.e., pallets and corrugated boxes, as separate and disconnected systems.

Despite the fact that the design of a unit load accounts for the interactions between both the pallet and stacked packaging, i.e., to ensure secure and safe transportation, the environmental implications of these two interacting unit load components have not been explored holistically. This study was undertaken in order to ensure a unit load design that optimizes for safety, cost-effectiveness, and reduced environmental impact by exploring the interactions and opportunities of diverse and interacting pallet and packaging material and design options. Building on the existing research into distribution packaging unit load optimization that is presented in Section 2, we describe our comparative scenario-based life-cycle analysis (LCA) methodology in Section 3, followed by results and discussion in Sections 4 and 5, respectively. This work (1) contributes new knowledge regarding how unit loads can be holistically and environmentally optimized based on the interactions between components, (2) demonstrates hotspots of environmental impact when optimizing unit load designs, and (3) provides thresholds for assessing the environmental advantages of the unit load optimization method proposed by Quesenberry et al. [7].

## 2. Research Background

Pallets are ubiquitous in the supply chain with 2.6 billion pallets circulating in the U.S. annually [8]. Out of all pallet materials, wood dominates in the industry with 94% of the market share [9]. Approximately 804 million wooden pallets were newly produced or recycled in 2016 from various industries in the United States [10]. According to the U.S. EPA, more than 11 million tons of EOL wood pallets were disposed via municipal solid waste (MSW) systems, with 3.1 million tons, or 27 percent, diverted to be recycled and 14 percent combusted for energy recovery [11]. In addition, the predominant type of packaging is the corrugated box, which accounts for 72% of the packaging materials used to build unit loads [12]. In 2018, corrugated boxes represented the single largest product category of generated EOL materials in the U.S., with 96.5% of the total 33.3 million tons generated being diverted for recycling [11].

The growth of e-commerce retail systems and an increased awareness of packaging waste has put pressure on industry members to reduce the environmental impacts of their operations [1]. Looking specifically at distribution systems, the environmental impacts of pallets have been explored in multiple LCA studies that compare the environmental impact of wooden pallets to the impact of pallets made from different materials, manufacturing systems, or alternative distribution packaging systems [13–17]. Other studies have developed life cycle inventories and investigated the environmental hotspots for wooden pallets in order to provide decision and design support for developing more environmentally friendly pallets [18–21]. Many studies across diverse research fields have assessed the environmental impact of corrugated boxes when they are used as part of a whole packaging system [22–26], exploration of EOL management options, and relative to emerging reusable packaging system alternatives as part of the circular economy [27–34].

However, given the magnitude of the packaging industry and its importance to commerce and economic growth, cost-effectiveness and the performance of unit loads also remain critical priorities for industry members. From an industry perspective, these must be balanced alongside environmental and sustainability concerns [31–34]. Accordingly, the design of the unit load—including considerations of both the pallet and package—provides a critical opportunity to consider diverse optimization considerations [35]. As with many other product categories, the pursuit of cost-effectiveness may result in the reduction of associated environmental impacts as well. In recent years, the effect of the stiffness

level of the wooden pallet top deck boards on the performance of corrugated boxes was broadly studied by researchers [7,35–37]. Quesenberry et al. [7] found that the stiffness of wooden pallets top deck boards affects the strength of corrugated boxes up to 37% when corrugated boxes are asymmetrically supported. They also found that this phenomenon can be used as a cost optimization method for unit loads by increasing the stiffness of the wooden pallet top deck boards and decreasing the board grade of corrugated boxes [7]. Kim et al. [35] further investigated this unit load cost optimization method and found that it can be affected by various unit load design factors like pallet wood species, top deck thickness of initial unit load scenario, corrugated box size, and board grade. Both studies clearly identified that this optimization method could be used to facilitate unit load cost optimization and may also provide a method for evaluating and strategically improving the environmental performance of current unit load designs at the same time.

## 3. Materials and Methods

This study mainly employed the life cycle analysis (LCA) method to compare the environmental impact of both the initial and the optimized unit load designs, which comply with international standard LCA guidelines ISO 14040:2006 and ISO 14044:2006 [38,39].

### 3.1. Goal and Scope Definition

The main aim of this study was to investigate the environmental effects of optimizing a unit load by increasing the stiffness of the pallets' top deck boards and reducing the board grade of its corrugated boxes using LCAs. The LCAs compared the environmental performance of multiple initial and optimized unit load scenarios with a cradle to grave perspective. This study included raw material production, packaging manufacturing, distribution, and end-of-life options (EOL) available via common municipal solid waste (MSW) operations. However, this study excluded the life cycle of packaged items from the system boundary due to their high variability. Figure 1 presents a drawing of the system boundaries considered in this study; the geographical scope was limited to the southeastern United States due to the variability of local wood species supplied for wood pallet manufacturing.

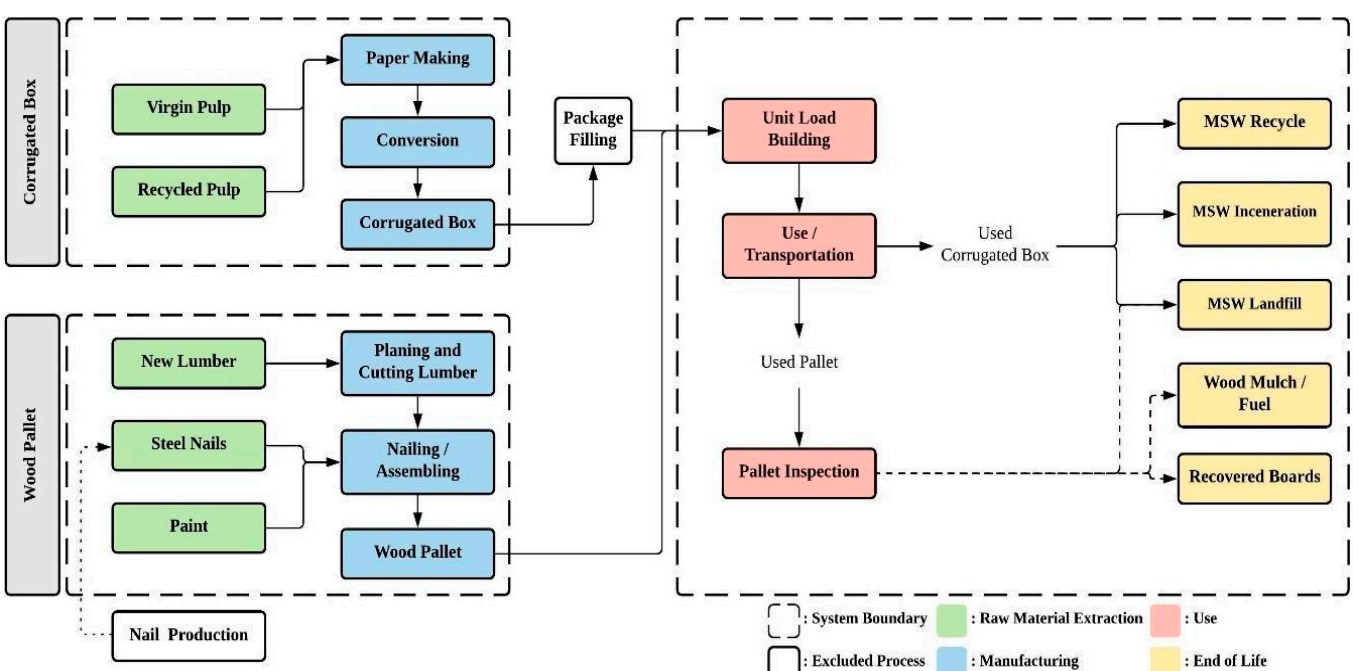

**Figure 1.** The schematization of the unit load system boundary.

The optimization method employed in this study required different amounts of pallet top deck material increase. It also resulted in different amounts of corrugated box material decrease, depending on various unit load design factors [30]. Hence, this study compared a wide range of paired initial and optimized unit load scenarios to investigate at what point they cross the line to show measurable environmental benefits through this unit load optimization method. Therefore, this study was not able to select specific load capacities for the functional unit. The functional unit for this study was defined as double-stacked unit loads with the same maximum safe load capacity under floor stacking conditions. These unit loads needed to be composed of a 1219.2 mm × 1016 mm Grocery Manufacturers Association (GMA) style, stringer-class, wooden pallet with identical corrugated boxes manufactured from the same flute-size corrugated board.

### 3.2. Unit Load Optimization Ratio and Scenario Analysis

In this study we compare the environmental performance of multiple pairs of initial unit load design (Figure 2a) and optimized unit load design (Figure 2b) scenarios composed of wooden pallets and corrugated boxes.

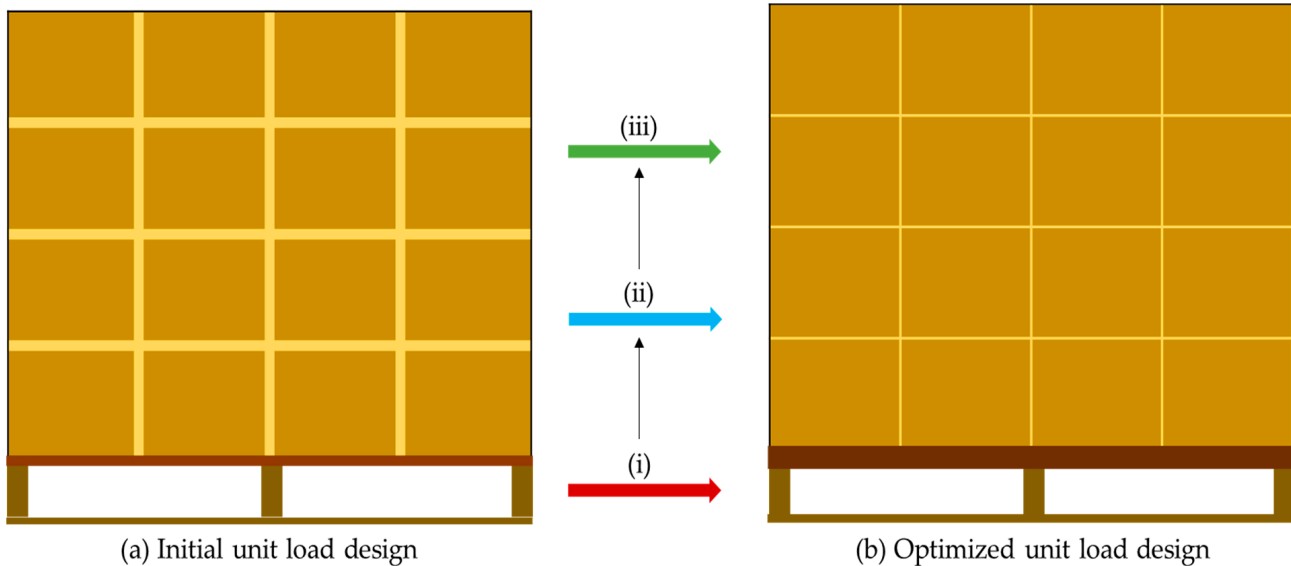

(a) Initial unit load design      (b) Optimized unit load design

**Figure 2.** Visual representation of the unit load optimization method proposed by Quesenberry et al. [7]. The method incorporates diverse design combinations in which (**i**) the pallet top deck stiffness is increased, leading to (**ii**) increased compression strength of the stacked corrugated boxes, and the resulting opportunity to (**iii**) reduce the board grade of the corrugated boxes while still achieving the same unit load compressive performance.

A total of 108 pairs of unit load scenarios were employed from the authors' previous study [35]. Kim et al. [35] designed an extensive list of common unit load design scenarios with varying pallet design factors, e.g., initial top deck board thickness and wood species, and varying package design factors, e.g., box size and board grade. The unit load scenarios were classified into three grades according to the amount that the top deck board thickness increased. Among these unit load scenarios, only scenarios with which it was feasible to apply the unit load optimization method from the manufacturing standpoint, grade 1 (less than 12.7 mm top deck thickness increase) and grade 2 (12.7 mm to 25.4 mm top deck thickness increase), were used for this study. The optimized unit load scenarios were created from the initial unit load scenarios by increasing the stiffness (thickness) of the pallet top deck boards and reducing the board grade of corrugated boxes while maintaining identical box performance (Figure 2). Detailed specifications of the investigated unit load scenarios were composed of green high-density hardwood (HD HW) pallets, green low-density hardwood (LD HW) pallets, green southern yellow pine

(GSYP) pallets, and kiln-dried southern yellow pine (KD SYP) pallets, and are described in the Tables S1–S4, respectively.

In order to effectively evaluate a wide range of unit load scenarios, four different pallet wood species groups were analyzed separately and ranked using a unit load optimization ratio (UOR) developed by the authors. UOR is a ratio of the decreased weight of corrugated materials to the increased weight of pallet materials, both of which change during this unit load design optimization process. Within this method, it is assumed that the weight of the corrugated board must decrease and the weight of the pallet materials must increase. The UOR was calculated by following Equation (1):

$$UOR = \frac{C_i - C_o}{P_o - P_i} \tag{1}$$

where:

*UOR* = Unit load optimization ratio
$C_i$ = Weights of corrugated boxes from initial unit load design
$C_o$ = Weights of corrugated boxes from optimized unit load design
$P_i$ = Weights of wooden pallets from initial unit load design
$P_o$ = Weights of wooden pallets from optimized unit load design.

Higher ratios reflect a smaller amount of weight increase in pallet materials to decrease in the weight of corrugated materials, which also can be assumed to be closer to a best case scenario. The worst case solutions have a lower ratio, which means there was a greater increase in the weight of pallet materials required to decrease the weight of corrugated materials. Unit load scenarios were ranked and listed in the order of material utilization according to the UOR of each scenario.

The weights of the wooden pallets and corrugated boxes from each pair of initial and optimized unit load scenarios were determined to calculate UOR. The weights of wooden pallets and corrugated boxes were computed by the industry-accepted pallet design software Pallet Design System™ (PDS™) v. 6.2 (National Wooden Pallet & Container Association, Alexandria, VA, USA) and the unit load design software Best Load™ v.4.0 (White & Company LLC, Blacksburg, VA, USA), respectively. Both software use finite element models to predict the performance of pallets and the packages shipped on pallets. Specifications of wooden pallets and corrugated boxes obtained from the previous study were entered into the corresponding software.

Once the unit load scenarios were ranked by UOR, six pairs of initial and optimized unit load scenarios from HD HW group, LD HW group, and GSYP group were selected for LCA based on regular interval ranking (e.g., first position, third position, fifth position). Exceptionally, all available unit load scenarios (only five pairs) were selected from the KD SYP group due to the limited number of optimizable scenarios. Among these scenarios, the unit load scenario with the lowest UOR from each wood species group was defined as the worst case scenario. And, the unit load scenario with the highest UOR was defined as the best case scenario in terms of proportional material utilization.

### 3.3. Environmental Performance and Life Cycle Analysis

The differences in environmental performance of the initial unit load designs and the optimized unit load designs from each of the scenarios selected at regular intervals for each wood species group were studied to investigate whether the unit load optimization method had environmental benefits or not. Positive environmental impact difference (+%) indicates that optimizing these unit load designs lowers environmental burdens, and on the contrary, negative environmental impact difference (−%) reflects the fact that optimizing these unit loads generates environmental burdens instead.

Life cycles of unit load scenarios were modeled through commonly used LCA software SimaPro 9.0 (PRe Consultants, Amersfoort, The Netherlands). The secondary inventory data regarding pallet weights were obtained from PDS™ v. 6.2. Unit load

design software Best Load™ v.4.0 was utilized to obtain the secondary inventory data of corrugated box weights. The U.S. LCI database, which traditionally represents U.S. regions, was mainly utilized for LCA modeling in SimaPro 9.0. The Ecoinvent v.3 database was also employed for LCA modeling to fill the existing inventory gaps within the U.S. LCI database.

### 3.3.1. Unit Load Raw Material Production

Inventory data regarding the raw material production for wooden pallets were modified from the most up-to-date life cycle inventory of wooden pallets in the United States, developed by Alanya-Rosenbaum et al. [20]. Values were adjusted according to the density of the four different wood species used in pallet manufacturing: green high-density hardwoods, green low-density hardwoods, green southern yellow pine, and kiln-dried southern yellow pine. The density of each wood species was calculated from PDS™ by dividing the weight (kg) of the pallets made from different wood species by each pallets volume ($m^3$). Inventory data were also broken down to the level of 'per 1 kg of wooden pallet raw material production' in order to be able to universally apply this data to the various unit load scenarios. Modified pallet raw materials production phase inventory data is listed in Tables S5–S8.

The inventory data for corrugated box raw material production were adopted from the comprehensive LCA study on the average corrugated products in the U.S. by the National Council for Air and Stream Improvement [36] due to its similar geographical scope to this study.

### 3.3.2. Unit Load Manufacture

For modeling the pallet assembly and corrugated box manufacturing, inventory data developed from the previous studies were also employed without modification since the input and output of data for unit load component manufacturing does not dramatically change due to unit load design factors. The average input and output data for the pallet manufacturing process were collected from Alanya-Rosenbaum et al. [20] due to their similar geographical scope to this study and the recentness of information. Inventory data for the corrugated box production process were obtained from the National Council for Air and Stream Improvement [36].

### 3.3.3. Transportation

The unit load that represents the functional unit for this study is a form of distribution packaging that serves multiple primary purposes including: to protect the products inside the packaging, and to distribute products more efficiently in bulk (vs. individually). Accordingly, the effective use phase for the unit load is during transportation throughout the supply chain. Thus, for purposes of clarity, we use the term "transportation", not "use", to refer to the movement of the unit load across distances using different modes.

Assumed transportation modes and travel distances for wooden pallets and corrugated boxes are presented in Table 1. Road transportation was mainly considered for wooden pallets due to this study's geographical scope, which also showed that wooden pallets are supplied and consumed locally. Only single use pallet scenarios were considered. Both road and rail distribution were considered for corrugated boxes since they could be supplied from anywhere in the United States. However, road transportation was regarded as the primary mode of distribution for corrugated boxes in order to follow the geographical scope of this study.

**Table 1.** Transportation data: distribution modes and distances considered for unit load components.

| Product | Item | Truck | | Rail | |
|---|---|---|---|---|---|
| | | Weighting (%) | Distance (km) | Weighting (%) | Distance (km) |
| Wooden pallet | Raw material to manufacturer | 100 | 148 | - | - |
| | Pallet manufacturer to product manufacturer | 100 | 148 | - | - |
| | Unit load transportation (use) | 100 | 1207 | - | - |
| | EOL transportation | 100 | 148 | - | - |
| Corrugated box | Wood logs to pulp and paper mills | 98.4 | 159 | 1.6 | 1577 |
| | Wood chips to pulp and paper mills | 94.5 | 299 | 5.5 | 1674 |
| | Recovered fiber to pulp and paper mills | 85.4 | 241 | 14.6 | 505 |
| | Pulp to pulp and paper mills | 80.1 | 262 | 19.8 | 1511 |
| | Chemicals | 72 | 217 | 28 | 1333 |
| | Containerboard to converting facility | 80.1 | 262 | 19.9 | 1511 |
| | Corrugated sheets to product manufacturer | 80.1 | 262 | 19.9 | 1511 |
| | Product to use | 95.7 | 283 | 4.3 | 2446 |
| | Unit load transportation (use) | 100 | 1207 | - | - |
| | EOL transportation | 87.4 | 241 | 12.6 | 505 |

### 3.3.4. End of Life of Unit Load

Unit load components' EOL phases were modeled based on the U.S. national data and details from the literature. At the end of a wooden pallet's life cycle, unbroken boards were recovered for reuse or repair of the other pallets, and the rest were landfilled, used for boiler fuel, or used for mulch and animal bedding [20]. Corrugated boxes end up being recycled in many cases, combusted for energy, or landfilled in fewer cases [2,4]. Table 2 presents the distribution of the EOL scenarios for each unit load component.

**Table 2.** End of life stage of unit load [2,4].

| | Wooden Pallets | Corrugated Boxes |
|---|---|---|
| Recovered boards | 37.3% | N/A |
| Fuel | 17.3% | N/A |
| Mulch and animal bedding | 40.4% | N/A |
| Landfill | 5% | 15.4% |
| Combustion | N/A | 3.7% |
| Recycle | N/A | 80.9% |

### 3.4. Life Cycle Impact Analysis

A midpoint-oriented life cycle impact analysis method, Tool for the Reduction and Assessment of Chemical and other Environmental Impacts 2.1 (TRACI 2.1), was utilized to calculate the environmental impact generated by the inputs and outputs of the unit load life cycle. This evaluation method was selected for calculations due to its matching geographical scope with this study. TRACI 2.1 was developed to reflect the environmental situation of U.S locations by the U.S. Environmental Protection Agency (EPA). A total of ten impact categories were calculated and presented including ozone depletion (kg CFC-11 eq); global warming (kg $CO_2$-eq.); smog (kg $O_3$-eq.); acidification (kg $SO_2$-eq.); eutrophication (kg N-eq.); carcinogens, measured in comparative toxic units for humans (CTUh); non-carcinogens (CTUh); respiratory effects (kg PM2.5 eq.); ecotoxicity, measured in comparative toxic units for aquatic ecosystems (CTUe); and fossil fuel depletion (MJ surplus).

### 3.5. Minimum Required Condition Analysis

The previous environmental differences analysis reported the crossing points between the negative and positive environmental impacts of the unit load optimization method. However, that analysis did not have a granular enough number of investigated scenarios to be able to find the exact first point where the positive environmental impact could be observed. Accordingly, a minimum required condition analysis was conducted where the finer steps around the crossing points from the previous analysis were investigated. The first unit load scenarios for each wood species group that showed an environmental benefit within all impact categories from the regular interval analysis were set as the base scenarios. The environmental performance difference analysis was repeated from the base scenarios to the scenarios that fell below 0% of the environmental performance difference (the break-even line) for any of the impact categories in descending order of the unit loads' rank for the minimum required condition analysis. The minimum required conditions to improve the environmental performance by optimizing unit load design were defined as UORs that will cause the unit loads to cross the environmental performance break-even line in all investigated impact categories for the first time.

### 3.6. Sensitivity Analysis

Sensitivity analysis on different ranges of transportation distance (during the use stage of the life cycle) where wide variation occurs was also conducted. The range of transportation distance is a parameter that can significantly vary for different reasons such as manufacturer's supply chain configuration and product sensitivity. Unit load scenarios with minimum required conditions for each wood species group were selected as base cases for sensitivity analysis. Two shorter and two longer ranges from the base cases of the range of transportation distances were investigated. Two shorter distances were 100 km and 500 km, while two longer distances were 1500 km and 2000 km, and the base case was set at a 1207 km range of transport distance.

## 4. Results and Discussion

### 4.1. Ranked Scenario Analysis Results

Figure 3 plotted the adopted unit load scenarios in ascending order of ranking based on the unit load optimization ratio (UOR) calculated for each wood species group. Columns colored red in Figure 3 indicate the unit load scenarios chosen at regular intervals to be utilized for the LCA. The best case scenarios were found to be unit load scenarios with a UOR of 3.200, 3.733, 4.480, and 1.918 for HD HW, LD HW, GSYP, and KD SYP, respectively. The worst case scenarios from HD HW, LD HW, GSYP, and KD SYP had a UOR of 0.035, 0.021, 0.023, and 0.16, respectively.

### 4.2. Environmental Performance Difference Analysis Results

Researchers compared and plotted the differences in environmental impacts between the initial unit load designs and the optimized unit load designs from each wood species group in order to investigate whether the unit load optimization method generated environmental benefit or environmental burden.

Figure 4 reports the environmental impact differences between initial unit load designs and optimized unit load designs within the HD HW group. It was observed that the unit load optimization method does not always create environmental benefits in the HD HW group. Optimized unit load design scenarios with a UOR of 0.035 generated up to 22.57% more negative environmental impacts than the initial unit load design in all impact categories except acidification. Although the unit load optimization method was not environmentally beneficial for the low UOR scenario, it started generating environmental benefits as the UOR increased. More than half of the impact categories (seven impact categories, excluding ozone depletion, eutrophication, and ecotoxicity) for the unit load scenario with a UOR of 0.189 showed environmental benefits by optimizing unit load design. Furthermore, it was discovered that optimizing unit load scenarios with a UOR



of 1.248 or higher improved environmental performance up to 22.93% in all investigated impact categories. Additionally, the unit load scenario with a 1.248 UOR was employed as the base scenario in the minimum required condition analysis. This was the first scenario that crossed the environmental performance difference break-even line during regular interval analysis.

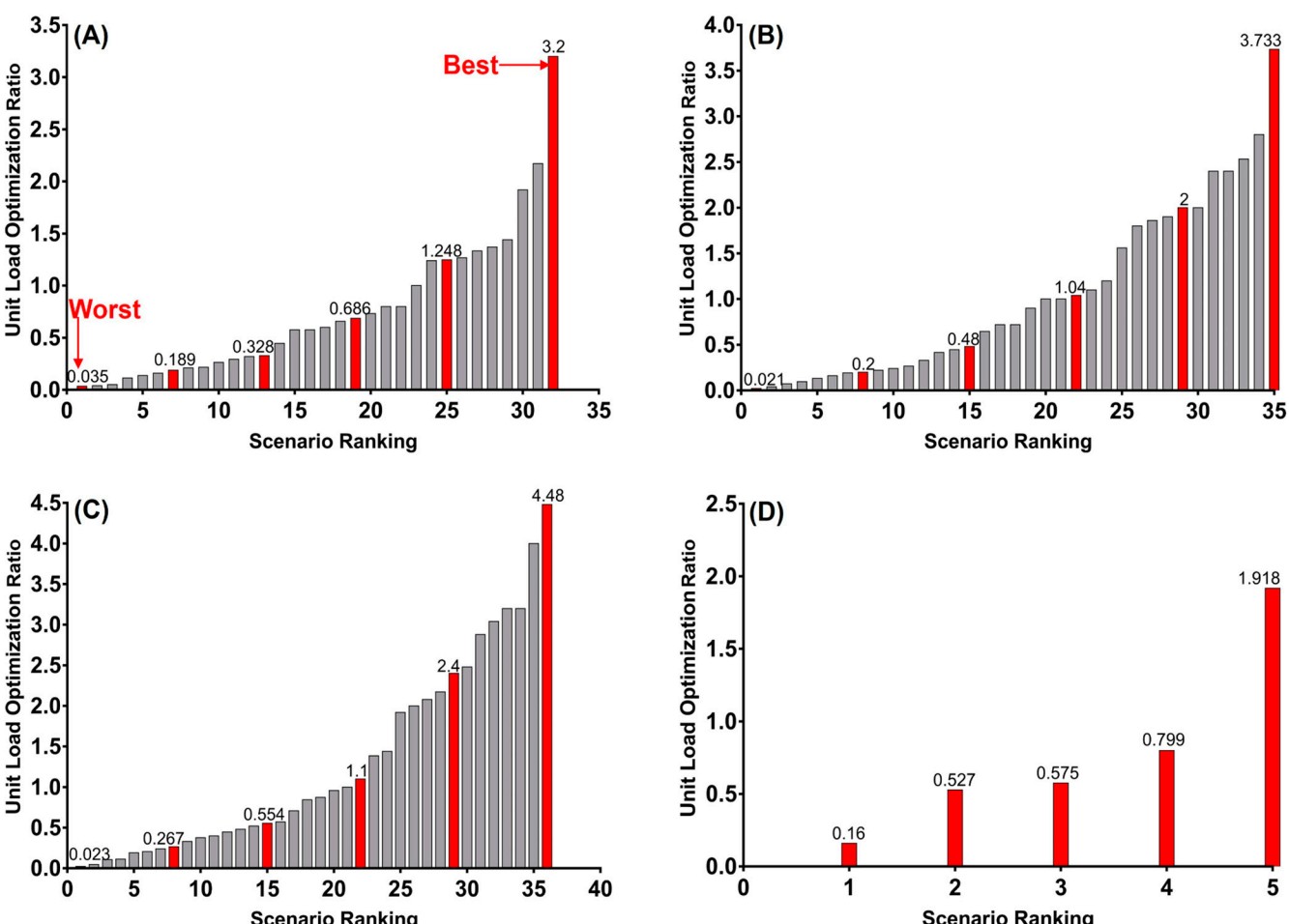

**Figure 3.** Rankings of investigated unit load scenarios from various wood species groups based on unit load optimization ratio. (**A**) Displays results of HD HW group, (**B**) displays results of LD HW group, (**C**) displays results of GSYP group, and (**D**) displays results of KD SYP group. Note: Red columns within plots indicate unit load scenarios selected for environmental analysis.

Figure 5 presents the environmental impact difference of the LD HW group's initial unit load designs and optimized unit load designs. Like the HD HW group, optimizing the first investigated unit load scenario with a UOR of 0.021 decreased environmental performance up to 35% in most impact categories except acidification. As the UOR increased, more and more impact categories started to show environmental benefits from the unit load optimization method. The unit load scenario with a UOR of 0.2 reported improvements in environmental performance from six impact categories, excluding ozone depletion, eutrophication, carcinogenics, and ecotoxicity. Moreover, all impact categories indicated that optimizing the unit load could generate environmental benefits as much as 22.85% within the LD HW group. The base scenario for minimum required condition analysis was set to 2 UOR in this case.

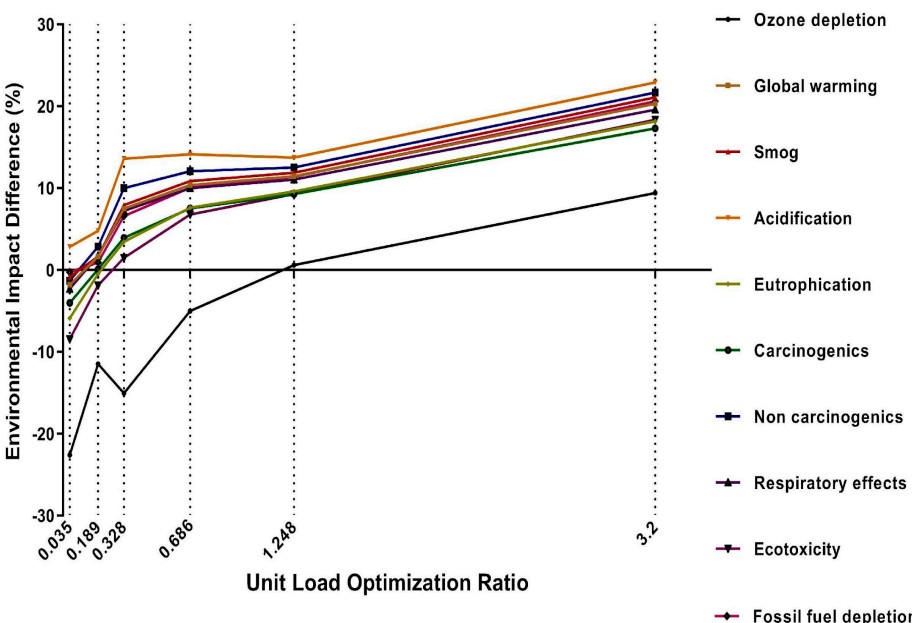

**Figure 4.** The environmental impact difference of initial and optimized unit load designs within the HD HW group.

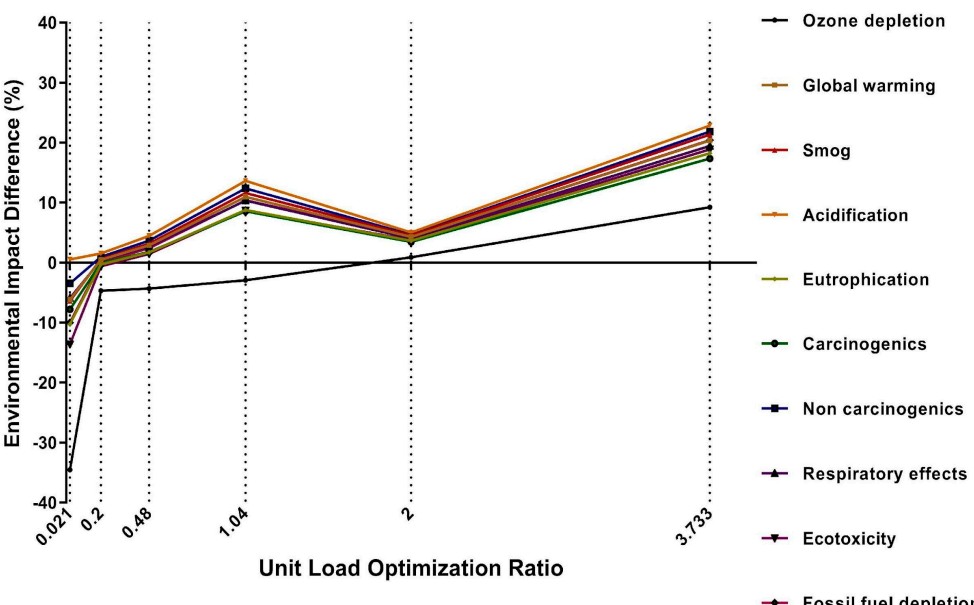

**Figure 5.** The environmental impact difference of initial and optimized unit load designs within the LD HW group.

Figure 6 displays the difference in the environmental performance of initial unit load designs versus the optimized unit load designs from the GSYP group. The trend of optimizing unit load scenarios with lower UORs causing diminishing environmental performance was also observed in the GSYP group. Optimizing the worst case scenario (0.023 UOR) decreased the overall environmental performance by as much as 35.23%, and an environmental benefit was only observed for acidification. GSYP group results also followed the trend discovered in the HD HW and LD HW groups where the number of impact categories showing environmental benefits from unit load optimization escalates as UOR increases. Consequently, the best case scenario in the GSYP group (4.48 UOR) showed as much as a 20.48% increase in environmental benefits in all impact categories when applying the unit load optimization method. There were environmental benefits in all impact categories when the unit load was designed with a UOR of 2.4 or higher in the GSYP group.

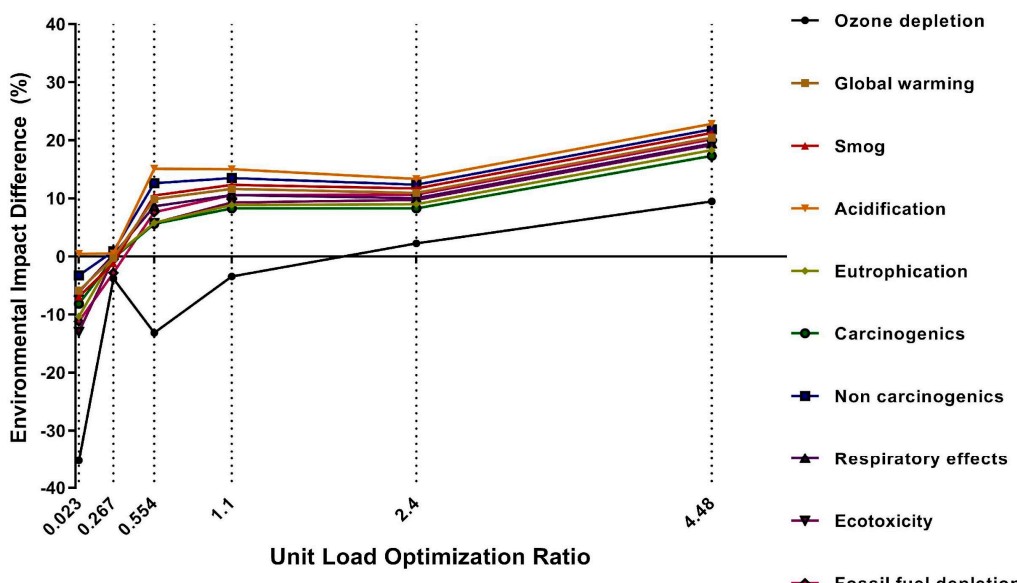

**Figure 6.** The environmental impact difference of initial and optimized unit load designs within the GSYP group.

Figure 7 shows the environmental impact difference of the initial and optimized unit load designs belonging to the KD SYP group. As the UOR escalates, improvement in the environmental performance through the optimization of the unit loads was also observed in many impact categories in the KD SYP group. The best case scenario improved the environmental impact of unit load up to 13.16%. However, the KD SYP group reported a slightly different trend from the other wood species groups. Previous results from the HD HW group, LD HW group, and GSYP group showed that optimizing the unit load can improve the environmental performance of unit load in all impact categories when the UOR is higher than specific points. In contrast, the KD SYP group could not enhance the environmental performance of unit load in terms of ozone depletion even with the maximum possible UOR. In other words, KD SYP unit load scenarios cannot expect full environmental benefits from this unit load optimization method.

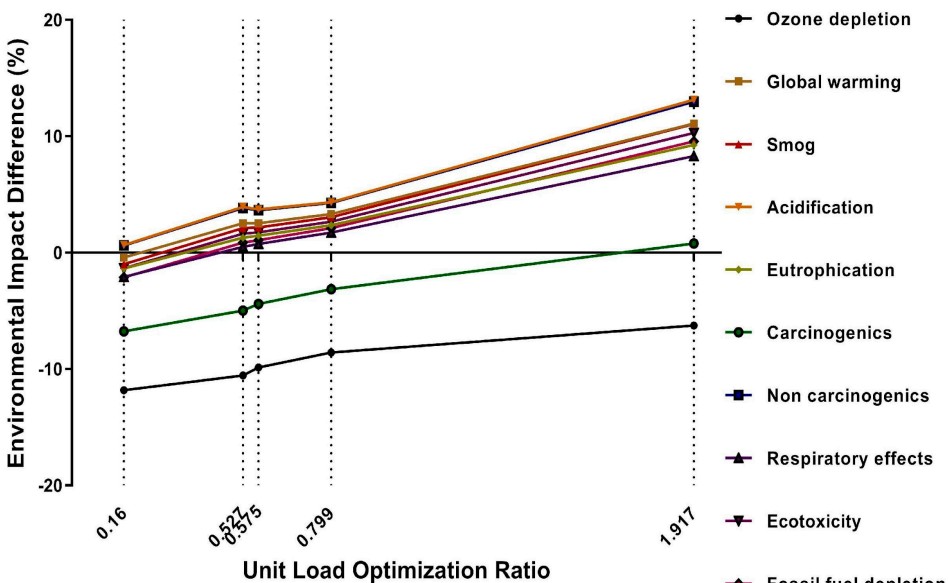

**Figure 7.** The environmental impact difference of initial and optimized unit load designs within the KD SYP group.

Overall, it was found that the investigated unit load optimization method could improve the environmental performance of unit loads when the unit load scenario has a specific UOR or higher. The first UORs for unit load scenarios that started showing environmental benefits in terms of all impact categories from the regular interval analysis were 1.248, 2, and 2.4 for the HD HW, LD HW, and GSYP groups, respectively. On the contrary, the KD SYP group showed only a low environmental performance increase in most impact categories after optimizing its unit load design. In addition, the KD SYP group could not achieve environmental benefits, in terms of ozone depletion, from any of the optimization scenarios. This exception was predominantly influenced by the limited number of KD SYP deck board sizes that can be effectively manufactured from the raw material. Therefore, the KD SYP scenarios investigated in this study required a significant increase in wood materials (only possible to increase from 11.1 mm to 17.5 mm) and there were not enough consecutive decreases in corrugated boards to offset the environmental burden of the unit load, even though the corrugated material requires more processing.

Although the KD SYP group results did not show environmental performance improvement in ozone depletion, the general trend of environmental benefits observed from more impact categories as the UOR increased remains unchanged within all investigated wood species groups. An increase of the UOR means that the unit load optimization process is requiring less of an increase in pallet wood materials proportional to the decrease in corrugated materials. In other words, a higher UOR utilizes less of a decrease in the chemically unprocessed wood materials compared to the chemically processed wood material increase during the unit load optimization process. This leads to impact categories that are heavily affected by the corrugated box-related phases which advances their environmental impact performance earlier than the impact categories that are less affected by the corrugated box-related phases. Further discussion on this matter can be found in the contribution analysis section below.

However, it was also discovered that unit load scenarios with a higher UOR do not necessitate greater environmental performance improvement. Fluctuations in the level of environmental impact differences were consistently observed regardless of the amount of UOR increase. This may be due to the fact that UORs were only based on the proportion of material utilization and do not account for the absolute amount of pallet and corrugated material change. In other words, unit load scenarios with lower UORs may have higher amounts of pallet and corrugated material changes during optimization than unit load scenarios starting with higher UORs. Since the UOR is only based on material efficiency, unit load scenarios with higher than or equal UORs to unit load scenarios that have crossed the break-even line for the first time can always expect environmental benefits by applying the unit load optimization method, but to different degrees. Therefore, the UOR that first crosses the environmental performance break-even line suggests the minimum required conditions to improve the environmental performance of the initial unit load design. These specific minimum required conditions for each wood species group were further investigated in the next section. It also means unit load designers need to run their own environmental performance difference analysis in order to figure the exact degree of environmental benefit received by applying the unit load optimization method to their specific unit load designs.

Figure 8 presents contribution analyses on six environmental impact categories: global warming potential, smog creation, fossil fuel depletion, eutrophication, acidification, and ozone depletion for each wood species groups' best case scenarios. The most significant contributors to global warming potential and acidification were the corrugated box raw materials production phases. Two leading contributors to smog creation and fossil fuel depletion were corrugated box raw materials production and transportation phases. Eutrophication was predominantly affected by corrugated raw materials production and box manufacturing processes. Ozone depletion was heavily affected by the pallet raw materials production phase, followed by the corrugated box raw materials production phase. Regarding the trend of ozone depletion, this trend was exceptionally difficult to generate a positive impact through the unit load optimization method, especially when compared to all other

impact categories. It required a relatively high UOR before observing any environmental benefit in terms of ozone depletion, and unit load scenarios in the KD SYP group were not able to create environmental benefit even with the high UOR. This was mainly because ozone depletion is governed by pallet raw material associated factors, mainly heating for the kiln dry process of lumber, while corrugated box-related processes controlled the other impact categories. This study's adopted unit load optimization method basically adds chemically unprocessed wood materials to the pallets top deck in order to decrease the chemically processed materials in the corrugated boxes. Consequently, environmental benefits in terms of ozone depletion, which is mainly governed by the pallet raw materials production phase, cannot easily be achieved until there is sufficient reduction of corrugated box raw materials phase to mitigate the effects of the pallet raw materials production phase.

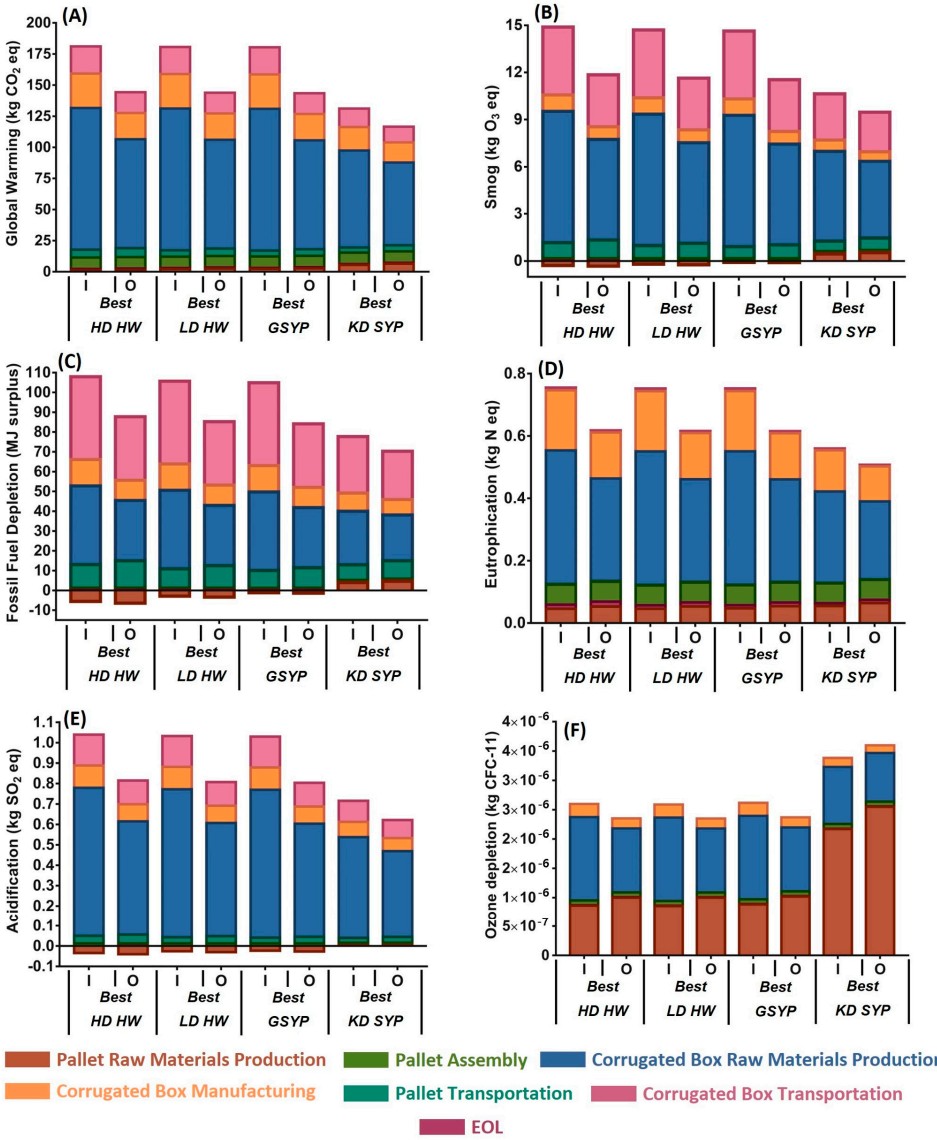

**Figure 8.** Contribution analyses for the best case scenarios from the HD HW, LD HW, GSYP, and KD SYP groups. (**A**) Shows global warming potential, (**B**) shows smog creation, (**C**) shows fossil fuel depletion, (**D**) shows eutrophication, (**E**) shows acidification, and (**F**) shows ozone depletion. I indicates initial case and O indicates optimized case.

### 4.3. Minimum Required Condition Analysis Results

This section estimates the minimum required conditions for each wood species group to estimate the environmental advantages obtained through the optimization of the unit

load designs. Figure 9 presents changes in the environmental performance of unit load scenarios: from the first unit load scenario that falls below the environmental performance break-even line to the first unit load scenario above that line during regular interval analysis for each wood species group. The results revealed that the UOR of 1.24, 1.56, and 1.92 are the minimum required conditions for the HD HW group, LD HW group, and GSYP group, respectively, to obtain environmental benefits in all impact categories by optimizing unit load. However, unit load designers need to look at these values as a preliminary decision-making tool, not as the exact values to use in deciding whether they should conduct an LCA on a specific unit load scenario. Many other minor factors can change their unit load's specific minimum required condition depending on the supply chain environment.

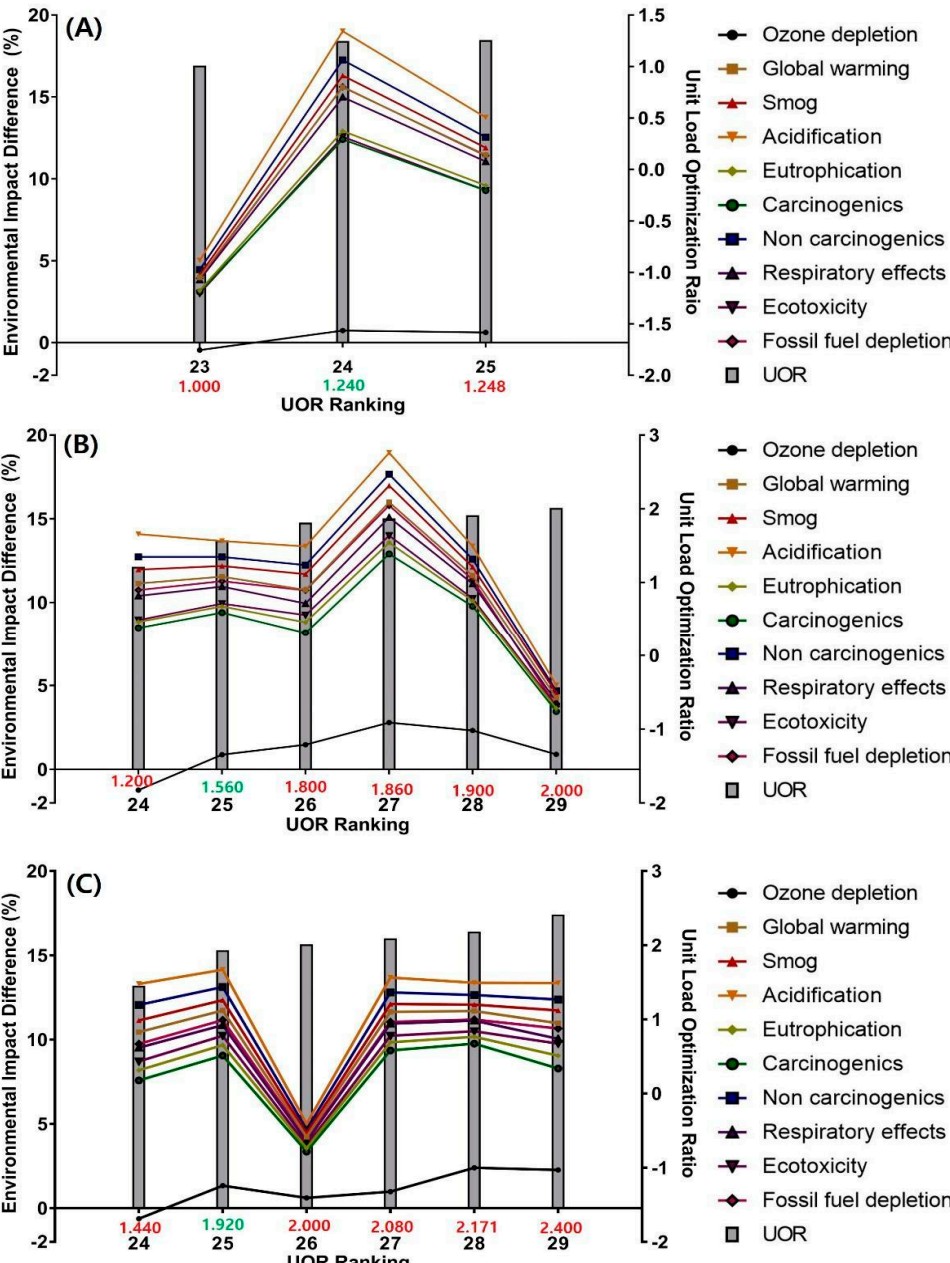

**Figure 9.** Environmental performance difference of unit load scenarios for minimum required condition analysis. (**A**) Shows results for HD HW group, (**B**) shows results for LD HW group, and (**C**) shows results for the GSYP group. Note: Red and green numbers report UOR of each unit load scenario. Green numbers especially indicate the minimum required condition of each wood species group.

As mentioned in the previous section, there are challenges in using UOR to directly predict the exact environmental impact of optimizing specific unit load scenarios. However, UOR still adds the value of easing the process of predicting the potential of environmental advantages of optimizing specific unit load design.

### 4.4. Sensitivity Analysis Results: Transportation

The impact of transportation distances on the use phase environmental performance of unit load scenarios with minimum required conditions for each wood species group are presented in Figure 10. Sensitivity analyses were conducted on unit load scenarios with the minimum required conditions by altering transportation distances, since these widely vary depending on the user. Increasing the transportation distance from 100 km to 2000 km resulted in decreasing environmental benefits from optimizing the unit load design as much as 6.69%, 3.43%, and 3.12% for the HD HW group, LD HW group, and GSYP group, respectively, in all of the impact categories except for ozone depletion. Ozone depletion did not show notable environmental performance changes nor dropped below the environmental performance break-even line for all three wood species groups. Considering ozone depletion as the most closely related impact category for determining the minimum required conditions, from the previous section, none of the impact categories were sufficient to change the environmental status of unit load scenarios with the minimum required conditions. Therefore, these sensitivity analysis results confirmed that the minimum required conditions do not change with different supply chain distance ranges.

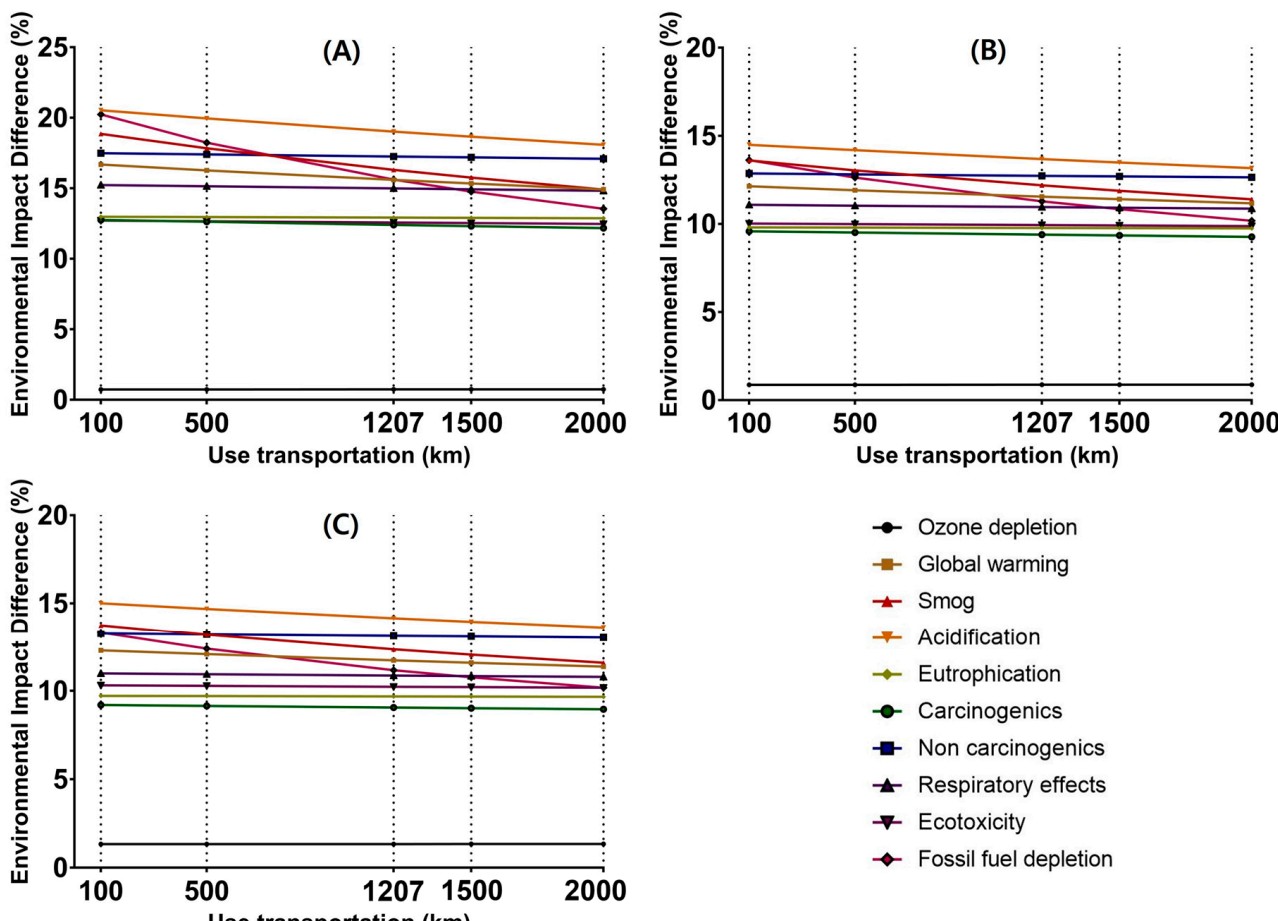

**Figure 10.** Environmental performance change of unit load optimization method as a function of different use phase transportation distances and wood species groups. (**A**) Presents results from HD HW group, (**B**) presents results from LD HW group, and (**C**) presents results from GSYP group.

## 5. Assumptions and Limitations

The following assumptions and limitations were made in this study:

1.  The modeling of UOR assumed scenarios in which the weight of the pallet was decreasing and the weight of the corrugated materials was increasing. This model assumes a desire for an optimized unit load on the basis of material use and performance. It is possible that all components could be increased in weight; however, such a design adjustment would contradict the intended objective of optimization.
2.  Employed unit load scenarios consisted of a 1219.2 mm × 1016 mm GMA style stringer class wooden pallet and three different sizes of corrugated boxes. Different styles or sizes of pallets and significantly smaller or larger sizes of corrugated boxes than the employed scenarios could affect the results.
3.  The wood species were limited to the four main species groups primarily used for pallet manufacturing in the southeastern United States. Different parts of the country use wood species that are easily accessible in their region to build pallets. The use of other wood species will affect the results due to the change in inventory data of pallet raw materials production phase.
4.  The LCA did not include the life cycles of the products contained within corrugated boxes and load stabilizers, such as a stretch wrap or band straps, due to their high variability.

## 6. Conclusions

This study was conducted to investigate the environmental implications of the unit load optimization method of increasing top deck board thicknesses and decreasing the board grade of corrugated boxes. Unit load optimization was able to improve the environmental performance of unit load design in many cases, e.g., by as much as 22.93%, 22.85%, 20.48%, and 13.16% for the high-density hardwood group (HD HW), low-density hardwood group (LD HW), green southern yellow pine group (GSYP), and kiln-dried southern yellow pine group (KD SYP), respectively. However, ozone depletion consistently showed a lower performance increase in all wood species groups. This was mainly observed in scenarios in which the required increase in pallet materials incurred greater ozone depletion impacts than the corresponding decrease in corrugated material was able to offset; ozone depletion is more heavily affected by the pallet-associated factors.

This study highlights and addresses several practical considerations for designers and packaging engineers. First, the UOR reflects the ratio of changes in pallet and corrugated box material utilization. A higher unit load optimization ratio (UOR) can, but does not, guarantee reduced environmental impacts. For example, unit load scenarios with a relatively low absolute quantity of pallet and corrugated box materials may have a higher UOR than scenarios with a relatively high absolute quantity of materials used. Therefore, it is highly recommended that designers conduct unit load design-specific environmental analysis in order to understand the actual degree of the environmental advantages gained by optimizing a particular unit load scenario.

Second, before proceeding to a complete LCA on a specific unit load design, unit load professionals should investigate the minimum required conditions to estimate whether there is the possibility of environmental benefits through unit load optimization. The minimum required UORs for the high-density hardwood group, low-density hardwood group, and green southern yellow pine group were reported as 1.24, 1.56, and 1.92, respectively. These values are guidelines and should only be used to support preliminary decision making regarding whether to proceed with further, more complicated analyses due to many other minor factors that could potentially affect them. A sensitivity analysis on use phase transportation confirmed that these minimum required UORs remain unchanged even if unit load users have unique use phase distances.

This work demonstrates that the unit load optimization method can be used to achieve environmental advantages alongside cost-effectiveness and performance parity. It also allows estimating the minimum conditions that must be present to generate environmentally

beneficial and cost-effective unit load designs. This study is the first to suggest to the packaging industry that distribution packaging can be environmentally improved by applying engineering knowledge of physical interactions between different levels of the packaging systems instead of developing whole new packaging systems or materials. Future work is needed to explore more diverse configurations of pallet and package designs and materials. Given the potential to dramatically support the sustainability transition of distribution systems, the use of digital technologies that integrate these methods—including environmental performance—should also be pursued.

**Supplementary Materials:** The following supporting information can be downloaded at: https://www.mdpi.com/article/10.3390/su151712687/s1, Table S1: Rank, UOR, and specification of investigated green high-density hardwood unit load scenarios for analysis [35]. Scenarios selected at regular intervals and minimum required condition analysis are highlighted as yellow and green, respectively; Table S2: Rank, UOR, and specification of investigated green low-density hardwood unit load scenarios for analysis [35]. Scenarios selected at regular intervals and minimum required condition analysis are highlighted as yellow and green, respectively; Table S3: Rank, UOR, and specification of investigated green southern yellow pine unit load scenarios for analysis [35]. Scenarios selected at regular intervals and minimum required condition analysis are highlighted as yellow and green, respectively; Table S4: Rank, UOR, and specification of investigated kiln-dried southern yellow pine unit load scenarios for analysis [35]. Scenarios selected at regular intervals are highlighted; Table S5: Inputs and outputs for the raw material production of 1 kg of wooden pallet built with green high-density hardwood (modified from Alanya-Rosenbaum et al. [20]); Table S6: Inputs and outputs for the raw material production of 1 kg of wooden pallet built with green low-density hardwood (modified from Alanya-Rosenbaum et al. [20]); Table S7: Inputs and outputs for the raw material production of 1 kg of wooden pallet built with green southern yellow pine (modified from Alanya-Rosenbaum et al. [20]); Table S8: Inputs and outputs for the raw material production of 1 kg of wooden pallet built with kiln-dried southern yellow pine (modified from Alanya-Rosenbaum et al. [20]).

**Author Contributions:** Conceptualization, S.K., L.H., J.P. and J.D.R.; methodology, S.K., L.H., J.P. and J.D.R.; software, S.K.; formal analysis, S.K.; writing—original draft preparation, S.K.; writing—review and editing, L.H., J.P. and J.D.R.; supervision, L.H.; project administration, L.H.; funding acquisition, L.H. All authors have read and agreed to the published version of the manuscript.

**Funding:** This research was funded by the Industrial Affiliate Program of the Center for Packaging and Unit Load Design at Virginia Tech.

**Institutional Review Board Statement:** Not applicable.

**Informed Consent Statement:** Not applicable.

**Data Availability Statement:** The data presented in this study are available in the Supplementary Materials.

**Acknowledgments:** The PDS™ software package used for data acquisition was provided by the National Wooden Pallet and Container Association. The Best Load™ software package used for data acquisition was provided by White and Company LLC.

**Conflicts of Interest:** The authors declare no conflict of interest.

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
