# Peer review of "Sustainable and Secure Transport: Achieving Environmental Impact Reductions by Optimizing Pallet-Package Strength Interactions during Transport"

_sustainability, doi:10.3390/su151712687_

Round 1

Reviewer 1 Report

In the research the cost optimization method of wooden pallets transporting stocked corrugated boxes is studied. The authors analyze the influence of many factors such as the stiffness of a pallet's top deck in order to establish how unit load optimization method could affect associated environment impacts. It was demonstrated that by reducing the corrugated board grade and increasing the the stiffness of the top decks, the overall environmental impact of transported unit loads can be reduced.

The article is written in correct and understandable language. However, a few corrections need to be done:

- page 1 line 36 and 37: "concern" is used twice;

- page 2 line 55: it should be "in the industry"

- page 2 line 79: "They" instead of "He"

- page 2 line 85: ...method could not...

- page 3 Fig. 1 - MSW need to be explained

- page 4 line 14: ...were described in Table as S1, S2, S3...

- please uniform the tense in the text e.g. page 4 line 14

- page 7 line 24: CO2 - 2 in subscript and so on, explain CTUh and CTUe

- page 8 Figure 2: instead of scenario ranking it should be better to use HD HW, LD HW and so on, respectively; there is a "t" missing in Ratio in OY title

- page 11 line 37: ....material requires more processing.

- page 14 line 44" to use UOR

- page 15 line 45: The impact of use transportation....

The article is written in correct and understandable language.

Author Response

Review #1

Comments

Response

The article is written in correct and understandable language. However, a few corrections need to be done:

Thank you for your feedback; we have addressed recommended changes

page 1 line 36 and 37: "concern" is used twice;

Thank you for noting this; we have modified the sentence to improve readability. 

- page 2 line 55: it should be "in the industry"

Thank you, this has been addressed.

- page 2 line 79: "They" instead of "He"

Thank you, this has been addressed.

page 2 line 85: ...method could not...

Thank you, this has been addressed.

- page 3 Fig. 1 - MSW need to be explained

Thank you - we have introduced the term and acronym in the preceding paragraph now.

- page 4 line 14: ...were described in Table as S1, S2, S3...

Thank you, this has been addressed.

- please uniform the tense in the text e.g. page 4 line 14

Thank you for noting this; we have done a comprehensive review and updated accordingly.

- page 7 line 24: CO2 - 2 in subscript and so on, explain CTUh and CTUe

Thank you - we have addressed both of these comments throughout the manuscript.

- page 8 Figure 2: instead of scenario ranking it should be better to use HD HW, LD HW and so on, respectively; there is a "t" missing in Ratio in OY title

The authors appreciate the reviewer’s comment. “t” is now added to y axis titles. However, the authors found keeping this figure showing scenario ranking is necessary for this ranked scenario analysis results to be used in the future sections. Results are separated by wood species using separate graphs for better readability.

- page 11 line 37: ....material requires more processing.

Thank you, this has been addressed.

- page 14 line 44" to use UOR

Thank you, this has been addressed.

page 15 line 45: The impact of use transportation....

Thank you for noting this; we have reviewed the entire document and revised this terminology accordingly.

Reviewer 2 Report

The paper presents a comprehensive analysis of unit load optimization and its impact on the environment. The study highlights the effectiveness of increasing the stiffness of the top decks and reducing the corrugated board grade in minimizing the overall environmental impact of transported unit loads. 

However, after careful consideration, it is my recommendation that this paper may not align closely enough with the scope and focus of the Journal. Additionally, it is advised to standardize the format of the figures. 

Minor editing of English language required.

Author Response

Review #2

However, after careful consideration, it is my recommendation that this paper may not align closely enough with the scope and focus of the Journal. Additionally, it is advised to standardize the format of the figures. 

We appreciate your time, consideration, and feedback. We believe that, given the goal of innovative approaches to reducing environmental impacts of transport packaging, this paper is aligned with the objectives of the Journal. However, we appreciate your comments and have modified the language to more appropriately emphasize the sustainability motivations and opportunities that are part of this work.  We have also adjusted the Figures to improve standardization and clarify.

Reviewer 3 Report

Thanks to the respected authors

The topic of the article is potential and necessary. The following suggestions are provided to improve the article:

1- In the introduction, the "necessity and purpose of the article" and the "questions and hypotheses of the article" must be mentioned directly.

2- The background of the research must be added to the article after the introduction. The article does not have a research background yet!!!

3- Use newer and more reliable sources in the research literature. The article needs reliable and new sources to be added in the body of the article, especially the research literature.

4- In the data analysis section, it is better to use images in addition to the graphs used to better understand the topic of the article.

5- In the conclusion section, practical results should be proposed - as well as suggestions for future research.

Good luck

Minor editing of English language required

Author Response

Review #3

The topic of the article is potential and necessary. The following suggestions are provided to improve the article:

We appreciate your time and feedback, and have responded to all your comments.

1- In the introduction, the "necessity and purpose of the article" and the "questions and hypotheses of the article" must be mentioned directly.

Thank you; we have revised the Introduction section and have more clearly stated the purpose and research questions now. 

2- The background of the research must be added to the article after the introduction. The article does not have a research background yet!!!

Thank you for this suggestion; we have separated and distinguished the literature review/background research section now, and have increased and updated the referenced literature.

3- Use newer and more reliable sources in the research literature. The article needs reliable and new sources to be added in the body of the article, especially the research literature.

Thank you, we have expanded the literature review and included additional and more recent references.

4- In the data analysis section, it is better to use images in addition to the graphs used to better understand the topic of the article.

Thank you, we have included a new figure (Figure 2) to help clarify the focus of the study. Given the technical nature of the analysis, it was not possible to include images of the resulting scenarios, e.g., differences would not be visible in such an image.

5- In the conclusion section, practical results should be proposed - as well as suggestions for future research.

Thank you, we have incorporated these suggestions.

Reviewer 4 Report

Dear authors, I would like to thank you for the rigor you have shown in your study titled "Defining the Environmental Performance of a Unit Load, Optimization Method for Increasing Deck Thickness on Pallet and Decreased Corrugated Cardboard Quality". The literature section is pretty self explanatory. References may need to be increased. The method section contains many sub-titles. These are likely to distract the reader. Organizing the method section with fewer subtitles can speed up reading. Results and data appear to be consistent.

Author Response

Review #4

Dear authors, I would like to thank you for the rigor you have shown in your study titled "Defining the Environmental Performance of a Unit Load, Optimization Method for Increasing Deck Thickness on Pallet and Decreased Corrugated Cardboard Quality". The literature section is pretty self explanatory. 

Thank you for your time, consideration, and helpful feedback.

References may need to be increased

Thank you - we have reviewed, updated, and increased our references throughout the manuscript.

The method section contains many sub-titles. These are likely to distract the reader. Organizing the method section with fewer subtitles can speed up reading.

Thank you for this suggestion; we have reduced the number of subtitles used in the Methods Section.

Results and data appear to be consistent.

Thank you for reviewing and confirming.

Round 2

Reviewer 2 Report

The revised manuscript already meets the publication standards of the journal. I recommend accepting it for publication.

Reviewer 3 Report

The changes and corrections of the article are approved

 Minor editing of English language required